# School Nurse Perspectives of Working with Children and Young People in the United Kingdom during the COVID-19 Pandemic: An Online Survey Study

**DOI:** 10.3390/ijerph20010481

**Published:** 2022-12-28

**Authors:** Dana Sammut, Georgia Cook, Julie Taylor, Tikki Harrold, Jane Appleton, Sarah Bekaert

**Affiliations:** 1School of Nursing and Midwifery, University of Birmingham, Birmingham B15 2TT, UK; 2Centre for Psychological Research, Oxford Brookes University, Oxford OX3 0BP, UK; 3Birmingham Women’s and Children’s Hospitals NHS Trust, Birmingham B4 6NH, UK; 4Oxford Health NHS Foundation Trust, Oxford OX4 4XN, UK; 5Formerly OxINMAHR (Oxford Institute of Nursing, Midwifery and Allied Health Research), Faculty of Health and Life Sciences, Oxford Brookes University, Oxford OX3 0FL, UK; 6Faculty of Health and Life Sciences, Oxford Brookes University, Oxford OX3 0FL, UK

**Keywords:** children’s health, COVID-19 pandemic, school health services, school nursing, social vulnerability, survey

## Abstract

School nurses are public health specialists with an integral role in the safeguarding of children and young people. This study gathered information about school nurses’ approaches to overcome practice restrictions as a result of COVID-19. A cross-sectional survey was administered to school nurses across the United Kingdom. Quantitative data were analysed descriptively. Qualitative data (free-text responses to open-ended questions) were analysed using reflexive thematic analysis. Seventy-eight participant responses were included in the analysis. Quantitative data highlighted increased workloads; decreased contact with service users; and difficulties in identifying safeguarding needs and working with known vulnerable children. Through qualitative data analysis, five themes were identified: a move from preventive to reactive school nursing; professional challenges of safeguarding in the digital context; the changing nature of inter-professional working; an increasing workload; and reduced visibility and representation of the child. The findings call for advocacy by policymakers and professional organisations representing school nurses to enable this professional group to lead in the evolving public health landscape; for commissioning that recognises the school nurse as a specialist public health practitioner; and for sufficient numbers of school nurses to respond to the emergent and ongoing health needs of children and young people.

## 1. Introduction

The COVID-19 pandemic prompted significant changes to the way school nurses delivered specialist public health services in the United Kingdom (UK). National lockdowns, school closures and widespread redeployment required school nurses to rapidly adapt their practice to ensure continued delivery of universal, personalised and specialist services. School nurses provide an essential link between schools and communities, working with school-aged children across state, independent and special education needs schools [1,2]. School nurses saw the focus of their work shift throughout the pandemic, with new COVID-19-related duties and escalating safeguarding caseloads consuming already-compromised time and resources.

## 2. Background

School nurses in the UK are registered public health nurses who work with children aged 5 to 19. The term ‘children’ throughout this paper will refer to any child or young person who would fall under the remit of a school nurse. Public Health England [3] and the School and Public Health Nurses Association [4] set out the role of the school nurse in three areas: universal, personalised or targeted, and specialist, with safeguarding forming a key component across all three (summarised in Appendix B, Box 0). Within this paper, the term ‘safeguarding’ refers to statutory and non-statutory child protection activity.

The UK government mandated a variety of lockdown measures during the COVID-19 pandemic, including a nationwide lockdown from March–May 2020, followed by a series of regional restrictions and subsequent lockdowns across England, Wales, Scotland and Northern Ireland. During lockdown periods, the majority of schools closed to all except the children of key workers and children who were considered vulnerable. Many school nurses faced the challenge of delivering health promotion and education services, alongside an increasing safeguarding remit, remotely or with significantly restricted access to school premises. In addition, the school nurse workforce was compromised due to sickness, redeployment and personal caring and/or home-schooling responsibilities. The impacts of lockdowns and COVID-19-related restrictions on school nurses’ work have been reported in other countries with similar school nursing models. Evidence from America [5], Hong Kong [6] and Sweden [7] alludes to an emerging duality of the school nurse role during the pandemic, with traditional focuses such as health promotion competing with new COVID-19 responsibilities.

The wider public health impacts of lockdown restrictions and reduced in-person contact for children have become apparent. Several international reports have highlighted children’s increased vulnerability during lockdowns to issues such as domestic abuse and exposure to substance misuse [8,9]. Online activity intensified, increasing children’s vulnerability to online bullying, emotional abuse, grooming, sexting and virtual sex [10]. Consequently, there was a significant increase in new child protection concerns [11]. For many children with pre-existing vulnerabilities and complex needs, the pandemic heightened exposure to risk, as respite and support became problematic [12,13].

Amidst these challenges, school nurses accelerated and implemented new innovative practices to maintain contact with children and their families; facilitate interprofessional working; and expand the reach of their services through extended partnership working. A recent scoping review of international literature highlighted a range of creative school nurse practices that were adopted to ensure continued and responsive services [14]. The review also highlighted that little research has been published exploring the experiences of school nurses during COVID-19, including in the UK. The aim of this survey study was to gather information on how school nurses’ work evolved as a result of the COVID-19 pandemic, with a focus on their work with vulnerable children and young people.

## 3. Materials and Methods

This cross-sectional study used an online survey that was developed to gather information on innovative practices in frontline school nurse work, focusing in part on work with vulnerable children, during the pandemic in the UK. Survey questions were informed by, and developed from, a range of interrelated policies and the research team’s professional practice and theoretical knowledge: for example, public health/school nursing models [4], family-centred care/partnership working [15], contextual safeguarding [16], and communication within safeguarding [17]. In addition, questions were further informed by scoping review findings and input from an expert advisory group (consisting of school nurses and professional organisation leads). Some questions were also adapted from a previous survey that was developed and implemented by a member of the research team [18]. Draft versions were reviewed by the team and advisory group and piloted on five Specialist Community and Public Health Nursing students at Oxford Brookes University. Iterative amendments were made based on feedback, mainly to improve question clarity, and the final survey was reviewed and approved by the research team and advisory group, and Oxford Brookes University Research Ethics Committee (registration no. 211550). The final survey included 20 main questions (all non-mandatory, some including embedded logic follow-up questions) across three sections (Appendix A). Survey questions generated both quantitative and qualitative data, and the survey concluded with an invitation to participate in a follow-up focus group or interview. This invitation requested contact details, but through a different platform to maintain survey anonymity.

The survey was live from 9 April 2022–31 May 2022. A convenience sampling approach was employed to reach UK-based school nurses, with an online survey advert being disseminated by professional school nurse organisations (School and Public Health Nurses Association, and Community Practitioners’ and Health Visitors’ Association) via their social media and internal communications channels. The research team also disseminated the survey via social media and other relevant professional networks. The advert contained an embedded anonymous link to the Qualtrics-hosted survey. The survey landing page included a brief description of the study, followed by the participant information sheet and an invitation to provide consent by checking a box. In order to maintain respondents’ anonymity, the survey was set to not capture or record IP addresses or geodata; as a result, multiple participation could not be ruled out. Participants could exit the survey at any time. Only fully completed and submitted surveys were included in the analysis; participants needed to select a final ‘submit’ option at the end of the survey for this criterion to be fulfilled. Only current practising school nurses (and school nurse students) based in the UK were eligible. Two hundred surveys were begun but after exclusion for non-completion/submission of the survey and not meeting inclusion criteria (e.g., non-UK-based), data from 78 participants were included in the analysis. This represents approximately 4% of the school nurse workforce in England and Wales [19]. The study was conducted and reported in line with the Consensus-Based Checklist for Reporting of Survey Studies (CROSS).

### Data Analysis

Quantitative data were analysed using SPSS 25.0 (IBM Corp., Armonk, NY, USA) [20] and are presented descriptively. Results were calculated excluding any missing data; where appropriate, a number of responses to an individual survey item are highlighted. Qualitative data from the open-ended survey questions were analysed using a six-step reflexive thematic analysis [21,22], involving: (1) familiarisation with the data; (2) generation of initial codes; (3) searching for themes; (4) reviewing themes; (5) defining and naming themes; and (6) writing the report. Across the seven open-ended questions, a total of 382 free-text responses were provided, totalling 9688 words. The exact breakdown for question responses was as follows: Q12.2 = 63 responses (1493 words); Q15.2 = 10 responses (187 words); Q16 = 71 responses (2116 words); Q18.2 = 58 responses (1541 words); Q19.2 = 62 responses (1113 words); Q19.3 = 64 responses (1274 words); and Q20 = 54 responses (1964 words) (see Appendix A). The data relating to each question were collated into separate transcripts, which were read and reread for familiarisation purposes. A semantic inductive coding approach was adopted, reflecting the descriptive nature of the data and a desire to foreground participants’ experiences. For example, the excerpts “I found contacting some families difficult as they chose not to engage at all” and “Others in the team had much more limited contact with children than would normally have been the case” were both coded as ‘reduced contact with service users’. Coding was undertaken by one researcher (S.B.) and reviewed and discussed by two additional researchers (D.S. and G.C.). A collaborative approach was taken to generate and refine themes, for which reflexive discussions were held to identify meaning patterns across the data and to describe the central organising concepts across code groupings [22]. The analytic findings were then reviewed by the advisory group and wider research team.

All members of the research team had a nursing or psychology registration, and all were involved in a wider project exploring global school nursing practices utilising multiple methodologies. The experience and knowledge obtained through this familiarity with the literature and school nursing practice more generally may have influenced the way survey data were viewed and analysed (for example, recognition of patterns of meaning based upon findings from earlier stages of the wider project). However, this would not have been to the detriment of the analytic process, but rather enabled findings to be located in the context of the team’s rich understanding of the issues facing school nurses during the pandemic.

## 4. Results

### 4.1. Descriptive Statistics

School nurses were represented across state, independent, mainstream and special education needs schools. The majority of school nurses in the sample reported working for state schools (88.6%). A large proportion of nurses reported working with primary (89.9%) and/or secondary (92.4%) aged children. The majority (53.2%) were members of a school nursing team who shared responsibility for schools in the area. The majority of participants were based in England (*n* = 69). However, all UK countries were represented: Scotland *n* = 6, Wales *n* = 1 and Northern Ireland *n* = 1 (1 unidentifiable postcode). See Figure 1 for a summary of participant locations. A range of working patterns were represented. Sixty-two (78.5%) held the Specialist Community and Public Health Nursing qualification. The sample represented an experienced school nursing cohort with participants reporting they had been working as school nurses for a mean of 12.35 years (SD = 8.69, min = 1, max = 39). See Table 1 for additional demographic details about the sample.

#### 4.1.1. Working during the Pandemic

Participants were specifically asked whether they were redeployed, experienced a change in workload, and/or a change in contact with children during the pandemic (see Table 2). The majority of the sample (84.8%) was not redeployed during the pandemic. Of those that were, this was for a mean of 10.4 weeks. Of the respondents who opted to provide further details (*n* = 11), all described placements in alternative community settings (i.e., outside of acute hospitals); for example, supporting district nursing and health visiting teams. The pandemic had a significant impact on school nurses’ workloads with 74.4% reporting an increase. However, the majority (60.3%) saw a reduction in their contact with children and their families during this time.

The modes of service delivery and communication methods employed by school nurses with children and their families and the wider multidisciplinary team were also impacted. All modes of alternative service delivery to face-to-face contacts increased (see Table 3). Telephone and online consultations along with short health promotional videos saw the biggest increase in use. However, only 11 (14.1%) participants reported that they had received feedback or conducted an evaluation of the school nursing services offered during COVID-19. All methods of communication aside from face-to-face meetings increased for methods of communication with the multidisciplinary team. Online, email and telephone communications increased in use the most.

#### 4.1.2. Working with Vulnerable Children

The majority of school nurses (86.1%) reported that COVID-19 restrictions impacted their ability to identify vulnerable children. Over three quarters (79.7%) reported their ability to work with vulnerable children and families known to them was negatively impacted (see Table 4). Nearly half of the participants (48.1%) reported that partnership working was harder; however, a notable proportion (21.5%) reported that this had improved.

### 4.2. Qualitative Findings

#### 4.2.1. A Move from Preventive to Reactive School Nursing

Participants noted that during the first lockdown, there was a lull in workload and then, with the successive return of children to schools (first key worker and vulnerable children, followed by full school returns), an increase in all types of referrals, predominantly safeguarding-related. Increased and sustained safeguarding work in response to these referrals radically changed the profile and focus of school nurses’ work, to the detriment of their wider role: “[School nursing was] dominated by safeguarding work rather than our public health role”.

Participants’ adaptability enabled them to maintain contact with the children on their caseload, respond to an increasing safeguarding remit, and to continue delivering health promotion and education: “We worked harder than ever to deliver our service in creative ways”. However, some participants noted that the required adaptability and flexibility were time-consuming: “Being flexible seemed to be required for everything, as even the simplest things were not straightforward due to COVID restriction[s]”.

Some participants commented that a lack of regular contact with vulnerable children in schools meant they were unable to undertake preventive or early intervention work. Consequently, issues once identified were greater: “We did not have eyes on children… Early identification opportunities were missed so problems, once uncovered, were much more complex”. The lack of time and opportunities for health promotion activities contributed to children having a health knowledge deficit: “A noticeable reduction in knowledge and understanding for sexual health related topics, for immunisation related understanding and mental health related topics”.

As a result of having to adopt an increasingly reactive rather than preventive approach, school nurses were faced with the consequences of increasingly risky sexual health behaviours and emergent mental health concerns among children: “The level of mental health support required [for children and young people] is increasing”. Participants reported how other support services changed their delivery model due to COVID-19-related challenges, again leaving school nurses to support children and families with acute and complex issues: “[High] expectations of what school nursing can realistically offer—with other services under extreme pressure, especially mental health services, too many times school nurses having to hold complex cases”. As schools closed, and school nurses faced restricted access to school grounds, early identification of potential health issues, and opportunities to mitigate negative outcomes, were missed: “Schools are a big protective factor for children and pick up a lot of needs. They weren’t able to do that, so once they were open we had an avalanche of referrals”.

#### 4.2.2. Professional Challenges of Safeguarding in the Digital Context

Participants highlighted particular challenges associated with using digital platforms for contact and communication, with some indicating that these were suboptimal for both informal and formal assessments: “I don’t feel a phone/WhatsApp contact is as effective as face-to-face contact at home or school, [you are] unable to do home environment assessment, observe parent/child interactions, non-verbal communication, etc.”. Participants alluded to an impeded ability to adequately gain and convey information about children’s circumstances: “Those times where we were unable to meet a child face-to-face has impacted the amount of information we are able to provide at multi-disciplinary meetings”. There was also a lack of visibility of children and their families to professionals in virtual statutory safeguarding meetings. One participant remarked: “I find constantly doing CPPs [Child Protection Plans] over [Microsoft] Teams a barrier. It is fine every now and then but we really should meet the parents in these environments. I don’t know who is in the room with them”. The way in which some parents or caregivers contributed to virtual meetings was a concern for some: “Core meetings moved to virtual ones—mostly off camera so we rarely saw the child/ren, parents/carers. As we were no longer seeing people face-to-face they were more reluctant to give information to us…[and might] minimise risks or hide abuse”. Some participants also voiced concerns about digital poverty, as not all families had devices that supported online modes of contact: “There was an inequality of the students we would be able to contact using remote methods, e.g., access to devices, internet and a space to speak in private”.

#### 4.2.3. The Changing Nature of Interprofessional Working

The majority of participants stated that multidisciplinary team meetings swiftly moved online and that this improved service communication: “Virtual social care meetings became more accessible as there was no travelling between meetings meaning we could attend more in a day”. There was also a significant increase in telephone and online communication among professionals, which many participants welcomed: “[It was] easier to contact other professionals as emails and other forms of communication were better shared or made available”; “For us, [there was] better sharing of information with schools, CAMHS [Children and Adolescent Mental Health Services], MHST [Mental Health Support Teams] and youth counselling services”.

Some key support services became more coordinated: “All services providing mental health support worked together to promote and support mental health”. New coordination strategies such as Early Help Network Meetings—introduced in one area to ensure a rapid coordinated response to concerns for specific children—were reported to be beneficial: “Work [is] more efficient now than it was before”. An additional benefit of changes to interprofessional working was a reported increase in appreciation for the school nurse role among colleagues: “I think my peers in the staff room have looked to me for leadership and advice and generally value my job and role more”; “The respect of what we do and our safeguarding knowledge and experience was valued”.

However, improved interprofessional working was not universally reported. Participants described various specific challenges associated with having to use virtual modes of communication. Many framed these challenges in terms of their own role and workload: “Meetings being called at short notice due to virtual nature”; “More meetings back-to-back, more clashes of meeting[s], harder to say no to meetings as no need to travel”. For some participants, partnership working became more challenging as it became difficult to contact members of the multidisciplinary team due to absence through sickness, redeployment, isolation and shielding: “Lack of contact with school staff. We were unable to go into school to speak to them. Trying to contact school staff by email was a lottery”. Where contact with the multidisciplinary team occurred, some noted a shift in focus to advising other services rather than supporting individual children: “Communication with school staff has often been to support them rather than to discuss pupils”. Participants also noted that obtaining information from some professionals was increasingly challenging: “As we were no longer seeing people face-to-face they were more reluctant to give information to us”. Further, the approaches and priorities of other professional groups and/or institutions were sometimes at odds with those of school nurses: “I didn’t feel my school’s approach to supporting vulnerable children was on a par with other schools. I was disappointed in them and this resulted in my having to have difficult conversations at times”. Some school nurses felt that the increasing reliance on virtual contact with professionals compromised their ability to build relationships and have informal discussions: “Loss of rapport, endless and overlapping MST [Microsoft Teams] meetings, no chance to have those ad hoc conversations…”.

#### 4.2.4. An Increasing Workload

In addition to the changing and extending nature of the school nurse role, the direct and indirect consequences of COVID-19 on the school nursing workforce were described frequently by participants: “High levels of staff absence meant we had to take on extra work when we were already at full capacity”. The impact of working within these conditions impacted the workforce in a multitude of ways: “Staffing redeployment, staff exhaustion, remote working disrupting team working and morale”; “New starters found it difficult to integrate into the team due to remote working causing staff retention issues”. The lack of in-person contact with colleagues also impacted staff in unanticipated ways: “I think we have underestimated the impact of not having face-to-face contact as colleagues, and perhaps not recognised before how much ad hoc supervision takes place when popping in and out of the office”. Online peer support and digital school nursing support forums were highlighted as being helpful in the rapidly changing COVID-19 landscape: “Facebook school nurse groups were invaluable”. The overall workforce pressures, alongside widespread redeployment, left some school nurses feeling that their service was undervalued: “The service was seen as dispensable and many staff were redeployed”.

#### 4.2.5. Reduced Visibility and Representation of the Child

While school-based drop-in services were unavailable, key in-person health delivery was not possible: “Routine visits, health education sessions, groups and workshops ceased during wave 1 [of COVID-19]”. School nurses adopted a range of alternative modes to facilitate contact with children such as ChatHealth (a confidential text messaging service), home visits or ‘walk-and-talks’. These services were variably introduced or foregrounded if already in place: “[We were] unable to see young people face-to-face due to not being in school; however, they made contact through our ChatHealth text messaging service which we promoted as much as possible”. Other adaptations and/or additions to service delivery were also described: “We developed a remote assessment tool, well-being interventions, eating/nutritional assessments, continence workshops, [we] had to quickly learn how to use Microsoft Teams”. Feedback from service users was reported to be positive: “Our well-being workshops were amended to virtual with positive feedback from children, young people and parent/carers who valued the support—at a time when emotional and mental health concerns increased”.

Despite these efforts, some participants indicated that processes became more complicated and resulted in reduced contact with service users: “[Children were] no longer based in school so access [was] much more difficult... [this] resulted in [a] decrease in contact initiated by YP [young people]”. Some participants also noted that children were less independent during the pandemic and that ensuring confidentiality during virtual contacts was a concern: “When texting/calling when they are at home [they are] more likely to be overheard, or explaining where they are going for walk and talks added a challenge”.

Even once schools opened up to onsite delivery, participants noted that it remained difficult to have contact with children. For example, some schools did not allow school nurses onsite: “Some schools would not let visitors into school, in spite of us having PPE [personal protective equipment]”. Finding spaces for in-person contact that complied with COVID-19 regulations was also a challenge: “Rules often dictated that distance was to be kept, doors opened, etc., and this was tricky, particularly when providing pastoral care”. Participants also highlighted the implications of wearing personal protective equipment on their ability to meaningfully engage with service users in-person: “Having to use personal protective equipment for face-to-face contacts reduced quality of interaction & trusting relationships [were] harder to build”, as well as the wider implications of this impeded ability to build relationships: “[We were] unable to gain the voice of the child as we lost the ability to meet with CYP [children and young people] and build safe confidential relationships”.

Some participants suggested that parents took advantage of COVID-19 restrictions to limit their children’s contact with school services: “Children [were] kept home more by families with COVID as an excuse so [we were] unable to access them due to not being in school, parents [were] not contactable and [we were] not able to home visit due to suspected COVID symptoms”. Where home visits were possible, many participants still suggested that the quality of their interactions with children and young people was compromised: “Having to see children at home meant they were less free to talk to us”. Moreover, participants noted that some specific groups were particularly difficult to access: “Many children from special schools and families were shielding due to being clinically vulnerable. Contact was often by phone”. Overall, participants were mindful of the loss of the voice of the child in the context of the pandemic: “They [children] became lost and invisible to professionals”.

## 5. Discussion

Quantitative and qualitative data obtained through this UK survey study illustrate the various ways school nurses’ practice was impacted by the pandemic, and findings support emerging evidence from other international research exploring the issue [14]. Nearly three-quarters of participants in the present study reported that their workload increased during COVID-19, and nearly two-thirds reported having reduced contact with children and families. All modes of alternative communication and service delivery, such as telephone and online meetings and/or consultations, were reported to have increased significantly, while face-to-face contact with service users decreased. Challenges in identifying and supporting vulnerable children were also reported. Qualitative analysis of data from open-ended survey questions yielded five themes which reflect the changes in school nurses’ practice during COVID-19: (1) A move from preventive to reactive school nursing; (2) Professional challenges of safeguarding in the digital context; (3) The changing nature of interprofessional working; (4) An increasing workload; and (5) Reduced visibility and representation of the child.

Safeguarding and the prevention of harm and ill health make up a central part of school nurses’ work. In a pre-pandemic qualitative study [23], school nurse participants from across England described what they considered to be dramatic changes to safeguarding practice in recent years, with one participant stating that safeguarding formed 90% of their role. The changing nature of safeguarding work was attributed in part to external circumstances, e.g., the increased prevalence of specific issues such as child criminal exploitation. However, participants also discussed the impacts of staffing shortages and commissioning arrangements, leading many to conclude that the focus of their work had shifted from primary to secondary prevention, i.e., preventing the recurrence rather than the emergence of safeguarding issues. These findings, published shortly before the emergence of COVID-19, mirror many of the conclusions reached in the present study—in particular, participants’ concern that reactive service provision is likely to continue due to the emergent consequences of COVID-19. Post-pandemic literature is evidencing an increase in certain issues for children such as emotional and behavioural problems [24], anxiety and depression [25], and impacted peer relationships [26].

It has been suggested that the pandemic and associated restrictions posed a ‘perfect storm’, with many parents under increased stress, and many children’s risk factors heightened alongside a reduction in typical protective services [27]. There is clear evidence that school nursing services were struggling prior to the pandemic, with a lack of staff and resources cited as barriers to school nurses’ ability to offer their fundamental service [23]. Many of these issues were exacerbated by COVID-19. Findings from the present study also suggest that the move from face-to-face to virtual service provision reduced the visibility of particularly vulnerable children. It is well acknowledged that the enduring impact of COVID-19 will continue to affect vulnerable children and families post-pandemic, thus underpinning the need for adequate safeguarding strategies [28].

Data from the present study illustrate that school nurses faced an increased workload within an already-diminished workforce, simultaneously dealing with redeployment and sickness. The volume of administrative work surrounding caseload management is a perennial concern for practitioners [29]; school nurses in this study similarly described increased casework arising from pandemic-related issues. As virtual modes of service delivery are likely to continue—particularly in the context of interprofessional working—it is important that standard operational procedures at a national and local level are reviewed to reflect this increasingly normalised mode of multidisciplinary communication. There should be clear directives on the length of meetings, the time between meetings, and reasonable time frames for preparatory and follow-up work. Looking at broader funding issues for school nurses prior to the pandemic, Dawe and Sealey [30] highlighted the challenge of trying to achieve the same outcomes with year-on-year decreases in the public health budget. This issue is not unique to the UK, and yet, without adequate investment in school health resources (including but not limited to staff), the consequences of the pandemic for vulnerable children and young people are likely to snowball.

Surveys administered by the Department of Education [31,32,33] during the pandemic illustrate variations in school referrals to children’s social care throughout the pandemic, correlating with school closures in the UK. For example, from May 2020 to July 2021, when compared to the same time period between 2017 and 2020, there was a general trend of reduced school referrals. In figures reported in the most recent survey [33], though the total number of referrals was comparable to previous years for the same period, school referrals had increased by 27%. Rather than reflecting an actual reduction and/or increase in safeguarding issues, these statistics likely reflect schools’ fluctuating ability to detect safeguarding needs. Supporting this, the majority of participants in the present study indicated that COVID-19 restrictions impacted their ability to identify vulnerable children.

There appears to be an enduring lack of value in the school nursing role, including a poor understanding of the role amongst many professionals [30]. Findings from the present study support this conclusion, with participants alluding to the fact that their role was perceived to be non-essential in the pandemic effort (e.g., redeployment) and, paradoxically, capable of absorbing the workload of other services (e.g., child mental health services). However, it also appears that, for some school nurses at least, undertaking their roles during COVID-19 enhanced their professional standing, improving their relationships and lines of communication with key professionals. The value of these enhanced relationships is likely ultimately to benefit the protection of vulnerable children. Perhaps countries with established school nursing models need to explore raising the knowledge of school nurses as valued public health professionals to best benefit the most vulnerable. In addition, evidence from America has demonstrated that school nursing services can provide extensive medical and productivity (parental and teacher) cost savings [34], which additionally supports the need for a fully functioning preventive service.

The reach and economy of online meetings is being realised in many work contexts and is forming part of a hybrid working model. This is in keeping with the digitisation of patient care recommended in the National Health Service (NHS) long-term plan [35], which recommends using technology to facilitate interprofessional communication and public access to care, as well as offering likely cost-saving benefits. The present study highlights that the pragmatic and necessary move to virtual/remote communication and service delivery with children is not without its limitations. Most notably these relate to concerns about confidentiality and the quality of interactions, as well as reduced visibility of the children themselves. That said, the increased use of digital platforms appears to have had distinct advantages for school nurses’ partnership working, and there is evidence from other clinical staff groups to describe the benefits of online multidisciplinary team meetings [36]. However, not all evidence on this matter is consistent, with a recent scoping review identifying numerous challenges to professional virtual working [14]. The present study further highlights that children were not always able to adequately engage with online services (for example, due to inconsistent access), with many participants indicating that online service provision was suboptimal for both staff and service users. It could be argued that the use of virtual/remote modes of communication and service delivery should supplement, rather than replace, face-to-face delivery post-pandemic. Future research should explore the effectiveness of delivering different aspects of the school nurse role virtually.

Perhaps the most important finding from this study was that despite the increased activity around trying to maintain ‘eyes on the child’, there was a perceived loss of the child’s voice in many processes throughout the pandemic. This adds a new dimension to findings from earlier work, which showcased a range of innovative practices that were accelerated or introduced by school nurses to maintain contact with children. The learning from this, and other models of school health provision such as remote school nursing [37], must be carefully considered to ensure that school health delivery best serves local children. The results of this national survey would suggest that digital platforms for communication and service delivery with children might be part of a possible toolkit for the dispensation of school nurses, and employed in relation to identified needs.

Practice, organisational and policy recommendations arising from analysis of data obtained through this survey study include:For professional organisations to continue to represent school nurses in relation to their changing work profile as a consequence of the pandemic. This will empower school nurses to negotiate the external expectations of their role;For governments and local authorities to recognise the value of the school nurse as a public health specialist by commissioning school health models that place experienced school nurses in leadership and coordination roles within school communities. These should be supported by a sufficient workforce to ensure effective preventive public health work;To recognise the strengths and limitations of virtual interprofessional meetings and utilise them accordingly (recognising that face-to-face meetings can be helpful for informal networking and discussion). This should be accompanied by clear directives on workload planning that recognise pre- and post-meeting work;To return to face-to-face contact with children and young people in health promotion, education and specialist work. This recognises the importance of building trust, ensuring confidentiality, and holistic assessment when working with children and young people;For local authorities to subscribe to a range of online/digital platforms that can form part of a toolkit for school nurses’ work with children and young people, employed according to assessed needs.

### Strengths and Limitations

This survey had 78 responses. The sample was self-selected and represented a relatively small proportion of UK school nurses. According to Launder [19], there were 2100 school nurses in England and Wales in 2019, therefore this survey’s sample size represents approximately 4% of the school nurse workforce across these two countries. (NB. This study covered all four of the UK nations; however, to the authors’ knowledge, there are no publicly available data describing school nurse numbers across the UK as a whole.) The small sample size limited the possibility of making inferences about complex relationships across the datasets. Additionally, the survey did not capture participants’ age or gender; it is, therefore, possible that recruitment methods for this study, which utilised social media and other digital communication strategies, may have disproportionately attracted younger participants. That being said, given the digital transitions necessitated by the pandemic, it is likely that the UK school nursing workforce as a whole is digitally competent. Moreover, the geographic range, various lengths of time in post, and the fact that all types of schools were represented suggest a good representation of the school nurse workforce in the UK. This diversity of representation, together with the contextual description provided by the qualitative data, strengthens the potential for findings to be applied to other contexts. The survey development was informed by a rigorous literature review, consultation with an advisory group, and piloting prior to finalisation, all of which served to maximise face validity.

As qualitative responses were gathered from open-text box survey responses, they were relatively short in length, and the research team was unable to probe or explore the nuances of key responses. The descriptive nature of the data resulted in a similarly descriptive approach to analysis. Future studies should utilise study designs that allow more in-depth exploration of school nurses’ experiences. This recommendation is being realised by the next stage of the wider project (to which the present study contributes), involving focus groups and interviews with school nurses.

Whilst this study specifically sought to capture the experiences of UK school nurses, it is likely that the findings are relevant to other countries with similar school nursing models, as well as other frontline professionals such as those working in the fields of social work, youth work and education. In addition, the key challenges of supporting vulnerable children are likely to be applicable to professionals globally who used similar service delivery models during, and in the aftermath of, the pandemic.

## 6. Conclusions

This UK-wide survey study captures practising school nurses’ experiences of working with children, young people and families during the pandemic, shining a light on the challenges they faced and the steps taken to overcome these. Despite school nurses’ apparent resilience throughout COVID-19, this study highlights many impacts of the pandemic that are likely to endure in its aftermath, both for school nurses and the communities they serve. School nurses’ increasing safeguarding remit, reduced contact with children and families, and the consequent shift towards reactive service provision are issues that are likely to self-perpetuate without action to address the wider challenges facing the workforce. There is a clear need for meaningful collaboration between policymakers and organisational stakeholders to engage with and give voice to frontline school nurses, many of whom have reported feeling undervalued and even forgotten post-pandemic. School nurses stepped up and adapted in challenging circumstances to continue to deliver their mandated service, and in doing so, facilitated the development and adoption of a range of new working modes and practices. School nurses’ roles and scope of practice changed drastically during COVID-19, with many of these changes likely to endure. The pandemic recovery period offers an opportunity for meaningful change to be enacted in a top-down approach, through understanding and evaluating the effectiveness of the different working practices adopted by school nurses during this period. Appropriate commissioning and resourcing will be integral to enable the continued reach of this invaluable service.

## Figures and Tables

**Figure 1 ijerph-20-00481-f001:**
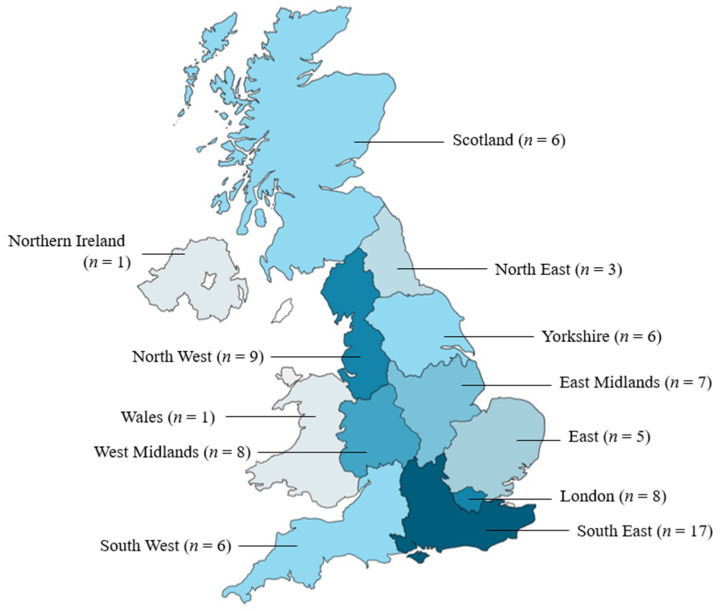
Geographical distribution of school nurse participants.

**Table 1 ijerph-20-00481-t001:** Description of the characteristics of the school nurse participants.

	Frequency	Percentage
**What type of school(s) do you work with? (all that apply)**		
State	70	88.6
Independent	14	17.7
Special needs	20	25.3
Other	3	3.8
**What age are the children you work with? (all that apply)**		
Primary school (ages 5–11)	71	89.9
Secondary school (ages 11–16)	73	92.4
Further education college	26	32.9
**Which of the statements best describes the way you work with schools? (all that apply)**		
I am attached to one school	7	8.9
I am attached to one school and I am the main school nurse	5	6.3
I am attached to named schools	7	8.9
I am attached to named schools and I am the main school nurse	29	36.7
I am part of a school nursing team who share responsibility for schools in the area	42	53.2
**What are your contracted hours as a school nurse?**		
Full time (all year)	29	36.7
Full time (term time only)	7	8.9
Part time (all year)	20	25.3
Part time (term time only)	19	24.1
Other	4	5.1

**Table 2 ijerph-20-00481-t002:** Broad work profile change for school nurses as a result of the COVID-19 pandemic.

	Frequency	Percentage
**Were you redeployed?**		
Yes	12	15.2
No	67	84.8
**Did your workload change during the COVID-19 pandemic?**		
No	7	9.0
Yes, it decreased	13	16.7
Yes, it increased	58	74.4
**Did you experience a change in children’s, young people’s or families’ contact with the school nursing service during COVID-19?**		
No change in contact	10	12.8
Decreased contact	47	60.3
Increased contact	21	26.9

**Table 3 ijerph-20-00481-t003:** The modes of service delivery used by school nurses to communicate with children, young people and families and the multidisciplinary team during the COVID-19 pandemic (reported as frequency and percentage).

	Increased	Same	Decreased	Never Used	Not Applicable
**The modes of service delivery used by school nurses to communicate with children, young people and families**					
Telephone consultations	72 (91.1)	5 (6.3)	1 (1.3)	1 (1.3)	0
Email *	54 (69.2)	17 (21.8)	4 (5.1)	3 (3.8)	0
Online/virtual consultations	70 (88.6)	3 (3.8)	1 (1.3)	5 (6.3)	0
Online classroom session *	28 (35.9)	2 (2.6)	10 (12.8)	29 (36.7)	9 (11.5)
Virtual nurses office **	27 (35.5)	4 (5.1)	3 (3.9)	31 (40.8)	11 (14.5)
Consultations outside, e.g., ‘Walk and Talk’ *	29 (37.2)	6 (7.7)	4 (5.1)	31 (39.7)	8 (10.3)
Short health promotion videos	43 (55.1)	5 (6.4)	1 (1.3)	23 (29.5)	6 (7.7)
Apps (such as ChatHealth)	36 (45.6)	14 (17.7)	3 (3.8)	20 (25.3)	6 (7.6)
Other ^	8 (29.6)	2 (7.4)	0	0	17 (63.0)
**The modes of communication used by school nurses to communicate with the multidisciplinary team**					
Telephone consultations *	63 (80.8)	13 (16.7)	2 (2.6)	0	0
Email	69 (87.3)	9 (11.4)	0	1 (1.3)	0
Texting/WhatsApp ***	39 (52.7)	18 (24.3)	1 (1.4)	14 (18.9)	2 (2.7)
Online/virtual meetings	78 (98.7)	1 (1.3)	0	0	0
Other ~	2 (11.8)	0	0	0	15 (88.2)

* Based on 78 responses, ** Based on 76 responses, *** Based on 74 responses, ^ Based on 27 responses, ~ Based on 17 responses.

**Table 4 ijerph-20-00481-t004:** The impact of COVID-19 and associated restrictions on school nurses’ ability to identify and work with vulnerable children, young people and families and the impact on partnership working.

	Frequency	Percentage
**Did COVID-19 restrictions impact your ability to identify vulnerable children, young people and families?**		
Yes	68	86.1
No	11	13.9
**Did COVID-19 restrictions impact your ability to provide support to vulnerable children, young people and families that were already known to you?**		
Yes	63	79.7
No	16	20.3
**Overall, considering the impact of lockdown and the resulting changes in workload, what has been the impact of COVID-19 on school nursing partnership working?**		
It improved	17	21.5
It stayed the same	9	11.4
It was harder	38	48.1
It was variable	15	19

## Data Availability

Due to the sensitive nature of the questions asked in this study, survey respondents were assured raw data would remain confidential and would not be shared.

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
