# Peer review of "School Nurse Perspectives of Working with Children and Young People in the United Kingdom during the COVID-19 Pandemic: An Online Survey Study"

_ijerph, 2022, doi:10.3390/ijerph20010481_

Round 1

Reviewer 1 Report

This is an interesting and important study on school nursing in the UK during the pandemic. Here are a few suggestions to improve the text:

1. Please put the study in a global context. A quick search online shows that there are several other studies about challenges for school nurses work during the pandemic around the world, see for example US (Hoke et al., 2021), Hong Kong (Lee et al., 2021) and Sweden ( Garmy et al. 2022).

2. Please provide a clearer information about the process of the qualitative analysis of the open questions.

3. Please provide information about how much text the qualitative data was based on (i.e. how many pages or how many words).

4. Please discuss the trustworthiness of the data, i.e reliability/validity for the quantitative data and trustworthiness for the qualitative data (for example credibility, dependability, transferability…) in the strengths and limitations section.

Reviewer 2 Report

Please, describe the sampling more precisely. It is difficult to describe such a method as purposive sampling, it is rather convenient sampling. The sample of 78 respondents is very small. Referring it to the number of school nurses combined in England and Wales is misleading, as there was one participant in Wales. In view of this, 4% may be not true. Nonetheless, geographic variation is an important point to emphasize. The sample characteristics lack age and gender (there is length of work at this position), which is a drawback of this tool. In many countries, school nurses are mainly older women. They may still have limited digital competence, which must have improved during the pandemic period. Reaching out to respondents through social media may have skewed the sample toward those younger than the average of this profession.

The survey was conducted online as a Qualtrics-hosted survey giving the opportunity to answer open-ended questions which is a qualitative part. The qualitative part increases the quality of the study. However, the method of analysis consisted of providing quotes without respondent ID which is a flaw in the study. It is unclear whether the quotes were selected biased or whether there were so few of these statements that all were given.  As a rule, survey participants are reluctant to answer open-ended questions online. It would be useful to provide information how many of the 78 persons completed at least one box with an open-ended question. Please include the number of open-ended questions included in the qualitative part in the questionnaire description.

The tabulations are limited to simple analyses of response frequencies without cross-tabulation. The group of nurses reassigned to another job, perhaps on the front lines of the fight against the pandemic, is puzzling. The workplace to which the individual was redeployed is not stated (question 10.3). It would be interesting to see if the fact of redeployment affects the pandemic workload assessment.

Describing the study's limitations related to sample size, the authors might add that it was difficult to study complex relationships having so few cases.

The conclusions are very general, and this section should be expanded It would be worthwhile to summarize the positive and negative consequences of working in a pandemic, from the perspective of the development of this professional group and the needs of children.

I am not a language expert, but some (abbreviated) phrases are glaring, such as the division of ages into primary and secondary (without the word school), or "am" instead of "I am" in the table. 

The method of citing literature like the APA system is inconsistent with the instructions for authors in IJERPH.

Round 2

Reviewer 2 Report

The manuscript has been improved in many points, and my comments have been comprehensively answered.